# Fibrotic Idiopathic Interstitial Lung Disease: The Molecular and Cellular Key Players

**DOI:** 10.3390/ijms22168952

**Published:** 2021-08-19

**Authors:** Anna Valeria Samarelli, Roberto Tonelli, Alessandro Marchioni, Giulia Bruzzi, Filippo Gozzi, Dario Andrisani, Ivana Castaniere, Linda Manicardi, Antonio Moretti, Luca Tabbì, Stefania Cerri, Bianca Beghè, Massimo Dominici, Enrico Clini

**Affiliations:** 1Laboratory of Cell Therapies and Respiratory Medicine, Department of Medical and Surgical Sciences for Children & Adults, University Hospital of Modena and Reggio Emilia, 41100 Modena, Italy; annavaleria.samarelli@unimore.it (A.V.S.); roberto.tonelli@me.com (R.T.); marchioni.alessandro@unimore.it (A.M.); giulibru92@gmail.com (G.B.); filippo.gozzi@unimore.it (F.G.); darioandrisani@libero.it (D.A.); ivana_castaniere@icloud.com (I.C.); linda.manicardi3@gmail.com (L.M.); antomor93@hotmail.it (A.M.); stefania.cerri@unimore.it (S.C.); bianca.beghe@unimore.it (B.B.); massimo.dominici@unimore.it (M.D.); 2Respiratory Diseases Unit, Department of Medical and Surgical Sciences, University Hospital of Modena, University of Modena Reggio Emilia, 41100 Modena, Italy; lucatabbi@gmail.com; 3Clinical and Experimental Medicine PhD Program, University of Modena Reggio Emilia, 41100 Modena, Italy; 4Oncology Unit, University Hospital of Modena, University of Modena and Reggio Emilia, 41100 Modena, Italy

**Keywords:** lung disease, idiopathic pulmonary fibrosis, myofibroblast, extracellular matrix proteins, tgfβ1 signalling

## Abstract

Interstitial lung diseases (ILDs) that are known as diffuse parenchymal lung diseases (DPLDs) lead to the damage of alveolar epithelium and lung parenchyma, culminating in inflammation and widespread fibrosis. ILDs that account for more than 200 different pathologies can be divided into two groups: ILDs that have a known cause and those where the cause is unknown, classified as idiopathic interstitial pneumonia (IIP). IIPs include idiopathic pulmonary fibrosis (IPF), non-specific interstitial pneumonia (NSIP), cryptogenic organizing pneumonia (COP) known also as bronchiolitis obliterans organizing pneumonia (BOOP), acute interstitial pneumonia (AIP), desquamative interstitial pneumonia (DIP), respiratory bronchiolitis-associated interstitial lung disease (RB-ILD), and lymphocytic interstitial pneumonia (LIP). In this review, our aim is to describe the pathogenic mechanisms that lead to the onset and progression of the different IIPs, starting from IPF as the most studied, in order to find both the common and standalone molecular and cellular key players among them. Finally, a deeper molecular and cellular characterization of different interstitial lung diseases without a known cause would contribute to giving a more accurate diagnosis to the patients, which would translate to a more effective treatment decision.

## 1. Introduction

Interstitial lung diseases (ILDs) are a heterogeneous group of pathologies that affect the lung parenchyma with wide inflammation and diffuse fibrosis [1]. Fibrotic ILDs cause the onset of progressive symptoms in patients that culminate in the decline of lung functions and often respiratory failure, which translates to a poor quality of life for patients. Interstitial lung diseases are a broad spectrum of lung pathologies that can be divided into two groups: ILDs with a known cause and ILDs without a known cause that give rise to the idiopathic form of pulmonary fibrosis [2,3]. Furthermore, the possible causes of the ILDs identified are represented by systemic diseases such as connective tissue disease [4,5] or environmental exposure to pneumotoxic drugs [6], radiation therapy [6], occupational exposures (e.g., asbestosis) [7] or allergens in the case of hypersensitivity pneumonitis [8]. The discrimination among different ILDs that present hetherogeneous inflammatory and fibrotic patterns even among patients with the same disease is critical for a more predictable prognosis and a more efficient management of the patients. Among ILDs, idiopathic pulmonary fibrosis (IPF) is the most common and severe, and for this reason the distinction between IPF and non-IPF ILDs is important given the poor prognosis in IPF compared to other fibrosing ILDs [9]. Idiopathic pulmonary fibrosis represents the most common form of idiopathic interstitial pneumonia (IIP) and is characterized by the progressive remodeling of the lung parenchyma structure, which exaggerates extracellular matrix deposition and irreversible scarring. IPF occurs primarily in older adults, with a median survival of 3 years after diagnosis, and it is diagnosed by clinicopathological criteria, including the radiographic and/or histological hallmark pattern of usual interstitial pneumonia (UIP) [10]. Prognosis remains extremely poor, since most patients die for progressive respiratory failure, often precipitated by acute events, namely, disease exacerbations [11]. Indeed, since the natural history and the course of the disease is variable among patients, the prognosis could be quite unpredictable [12]. Despite the recent introduction of the two antifibrotic drugs, namely, pirfenidone and nintedanib, that can slow down the respiratory functional decline of IPF patients according to both the real-word data and randomized controlled trials, such as CAPACITY and ASCEND, where they improved survival in patients, IPF still has a high mortality rate and survival times are quite heterogenous [3,13,14,15]. Besides the IPF, the IIPs include the non-specific interstitial pneumonia (NSIP), which is an interstitial lung disease that may be both idiopathic and secondary to connective tissue disease, toxins or other causes [16], cryptogenic organizing pneumonia (COP), also known as bronchiolitis obliterans organizing pneumonia (BOOP), that has been hypothesized to be secondary to alveolar epithelial injury due to an unknown insult, acute interstitial pneumonia (AIP) that is an extremely severe idiopathic acute interstitial disease [17], desquamative interstitial pneumonia (DIP), which is strongly associated with smoking, respiratory bronchiolitis-associated interstitial lung disease (RB-ILD), characterized by the combination of interstitial disease and respiratory bronchiolitis, and lymphocytic interstitial pneumonia (LIP) that in the majority of patients is associated with systemic autoimmune or immunodeficiency disorders, including connective tissue diseases, or in rare cases can be idiopathic [18]. The most updated version of the guidelines concerning the IIPs from the American Thoracic Society/European Respiratory Society (2013) [19] introduced some changes regarding the classification of these diseases compared to the original classification (2002) [20]. In particular, the IIPs are divided into four main groups (chronic fibrosing, smoking-related, acute/subacute, and rare) with the addition of a new disease: idiopathic pleuro parenchymal fibroelastosis [1]. Most of the ILDs have unknown etiology while there are a group of ILDs related/due to occupational and environmental exposure [21,22,23]. Although the majority of the ILDs can have unknown etiology, different works suggest that environmental factors could represent a risk factor in ILDs such as IPF since they may increase the probability of developing the disease in genetically susceptible individuals. In particular, according to the current integral model, the fibrotic pathway in IPF patients is activated by recurrent alveolar epithelium injury (e.g., environmental risks) that, together with compromised repair mechanisms of alveolar epithelium, leads to the initiation, development and progression of the disease [24,25,26]. Thus, the alveolar epithelial cells (AECs) are unable to properly respond to repetitive injuries, resulting in the loss of epithelial integrity that, together with the secretion of pro-fibrotic factors, represents the initial crucial mechanism of IPF development promoting fibroblast migration, proliferation, activation and differentiation into myofibroblasts with deposition of Extracellular Matrix (ECM) and the following distortion of the lung architecture. From a macroscopic point of view, these processes cause an increase in the stiffness of the pulmonary parenchyma, leading to the impairment of gas exchange in the alveolar district. Since the epithelial cells represent the initiator of the pulmonary fibrosis and the myofibroblasts are the key effector cells that orchestrate the progression of the disease, several recent studies aiming to clarify the molecular mechanism behind the IPF progression are focused on the identification of the cellular origin of myofibroblast. Initially, different works connected the cellular identity for the myofibroblasts origin to resident lung fibroblasts able to directly differentiate into myofibroblasts under profibrotic stimuli [27], epithelial cells undergoing mesenchymal transition, namely, epithelial–mesenchymal transition (EMT) [28,29], bone marrow (BM)-derived cells as circulating fibrocytes [30] and pericytes [31,32]. Lately, the contribution of the epithelial cells through the EMT and the circulating fibrocytes have been objects of several debates [33,34] to such an extent that they are not considered to be cells giving rise to myofibroblasts. To date, given the results obtained from cell-lineage tracing experiments using reporter mouse models, four different cellular types have been demonstrated to acquire the myofibroblastic phenotype [35]. Among them, we found the following: interstitial lung fibroblasts localized in the interstitium immediately adjacent to alveolar epithelial cells [33], lipofibroblasts located near the AECs are lipid-droplet-containing interstitial fibroblasts [35], pericytes within the capillary basement membrane [36], mesothelial cells from pleural-mesothelium that line the visceral and parietal pleural surfaces [37], and resident lung mesenchymal progenitors, whose contribution in the myofibroblast-differentiating process seems prevalent [38]. Since it has been demonstrated that non-IPF ILD may display a progressive phenotype as IPF, opening the possibility to explore the use of antifibrotic drugs for the non-IPF patients, and given the positive outcomes from this, we could speculate that different ILDs may share a similar molecular mechanism that culminates in the fibrogenic phenotype [39,40]. Indeed, since the interstitial lung disease can include rare, heterogeneous and poorly understood diseases, their classification, the identification of the different subtypes, an accurate diagnosis and the prediction of disease progression could be challenging for the clinicians [41]. The purpose of this review is to summarize the state of the art of the key molecular mechanisms that characterize the onset and progression of the IIPs with a particular focus on the molecular scenario characterizing the fibrotic lung in IPF patients. The identification of multiple mechanisms behind the proliferation of resident fibroblasts, the transdifferentiation of the above-mentioned cellular type in myofibroblasts and the aberrant deposition of the extracellular matrix in the fibrotic lungs could be crucial for the identification of new therapies that would be able to reverse the pathological mechanisms.

## 2. Diagnosis and Histopathological Pattern of IIPs

From a diagnostic point of view, IPF is histologically identified by as usual interstitial pneumonia pattern (UIP) [42], characterized by a heterogenous phenotype with the presence of fibroblastic foci areas interspersed both with an area where the parenchymal lung is mostly preserved and with an area of inflammation and honeycombing. The High-resolution computed tomography (HCRT) of IPF patients shows a peripheral, subpleural and basal pattern of distribution characterized by reticular opacities, honeycombing, minimal ground-glass opacity and architectural distortion [17]. Indeed, the updated version of the American Thoracic Society/European Respiratory Society (ATS/ERS) [19] in 2018 stated an additional recommendation for a multidisciplinary discussion for diagnostic decision making. It occurs that in some cases of interstitial lung disease where there is clinical suspicion of IPF, the HCRT pattern might suggest an alternative diagnosis. Among such cases, nonspecific interstitial pneumonia (NSIP) represents the second most common type of IIP after IPF, accounting for 25% of IIP cases [43]. Furthermore, non-specific interstitial pneumonia (NSIP) can be either idiopathic or associated with connective tissue diseases (e.g., scleroderma), hypersensitivity pneumonia, drug reactions, or diffuse alveolar damage (DAD) and affects more often women ranging from 40 to 50 years of age [23]. The histological pattern of NSIP is characterized by the homogeneous inflammation and expansion of alveolar walls; there are three different subtypes depending on the pattern distribution—cellular, fibrotic and mixed—that are related to different prognoses, where the fibrotic is the representative for the worst prognosis. The HCRT of NSIP patients shows both similarity with IPF patients and peculiar findings that allow for discrimination from radiologists such as extensive ground-glass opacity and the presence of traction bronchiectasis with subpleural sparing in fibrotic non-specific interstitial pneumonia [19]. Indeed, cryptogenic organizing pneumonia (COP), also known as bronchiolitis obliterans organizing pneumonia (BOOP), is mostly found as secondary to connective tissue diseases, drug-induced adverse pulmonary reactions, hypersensitivity pneumonia and infectious processes. It occurs in patients between 50 and 70 years of age, without gender preference, and from the histological point of view it is characterized by organized buds of granulation tissue within the alveolar ducts and adjacent alveoli that create the obstruction of the alveolar lumen and bronchioles culminating in respiratory failure [21,42]. Acute interstitial pneumonia (AIP) is a severe form of idiopathic acute interstitial disease affecting patients between 50 and 60 years of age associated to a poor prognosis with a mortality rate greater than 50%. Although the etiology is unknown, a case report was published that stated how acute interstitial pneumonia can be triggered by strenuous exercise [44]. In general, the clinical, radiological, and histological patterns are similar to those of acute respiratory distress syndrome (ARDS). Typically, the histopathological pattern of AIP is characterized by diffuse alveolar damage (DAD) similar to the pattern found in ARDS [1] with the presence of the edema in the interstitium and alveolus during the early phase of lung injury, followed by the fibroblastic proliferation and type 2 cell hyperplasia in the organizing or late phase [45]. The HRCT shows ground-glass opacities and air space consolidation together with traction bronchiectasis indicating progression from the exudative phase to the proliferative fibrotic phase showing a strong correlation with the different phases of DAD [45]. Desquamative interstitial pneumonia (DIP), which is also a rare pathology mostly affecting men between 30 and 50 years, is strongly related to cigarette smoking and in some case is associated either with occupational risk exposure or the use of inhalation drugs. From a histopathological point of view, DIP is characterized by the thickening of the alveolar septa and massive alveolar infiltration of macrophages leading to interstitial inflammation and fibrosis. The HCRT shows bilateral ground-glass opacities with lower lobe predominance [46]. Lymphocytic interstitial pneumonia (LIP) that is part of the spectrum of benign pulmonary lymphoproliferative disease is characterized by the presence of the lymphocytic infiltrate that expand and culminate into the alveolar region. Furthermore, LIP has been associated with several autoimmune disorders (e.g., Sjögren’s syndrome, rheumatoid arthritis, systemic lupus erythematosus) [47], other pathological conditions such as dysgammaglobulinemia, infections (human immunodeficiency virus and Epstein–Barr virus), and genetic predisposition. The HCRT displays thin-walled cysts in a random distribution, along with ground-glass opacities, centrilobular and subpleural nodules and reticulonodular opacities [18]. Indeed, respiratory bronchiolitis-associated interstitial lung disease (RB-ILD) is a rare, inflammatory pulmonary disorder that mostly affects smokers between 30 and 60 years of age without gender preference. Similar to DIP, RB-ILDs are characterized by the presence of pigmented macrophages within the lumens of respiratory bronchioles and alveolar ducts. The HRCT of RB-ILD patients reveals central and peripheral bronchial wall thickening, centrilobular nodules, and ground-glass opacities together in some case with centrilobular emphysema at the upper lobe. Idiopathic pleuroparenchymal fibroelastosis (PPFE) that has been recently included in the ATS/ERS among the IIPs classification is characterized by the presence of fibrosis in the pleural surfaces and subpleural parenchymal lung, particularly located at the upper lobe. [48] Furthermore, PPFE that mostly occurs in males and with unknown cause [49] has also been reported as a consequence of bone marrow transplantation as chronic graft-versus-host disease [50]. Then, the HRCT of PPFE patients shows pleural thickening in the upper pulmonary zones together with subpleural reticulations in the upper and middle regions and bronchiectasis [49]. Finally, the diagnostic approach through the evaluation of HCRT offers a crucial tool for the diagnosis of IIPs since some radiological patterns specifically and particularly characterize different IIPs (Figure 1). Despite this, sometimes the correct and early diagnosis based on HCRT findings is challenging, especially when the IIPs are represented by heterogeneous patterns shared among/overlapped with other different IIPs as they are for the heterogeneous radiological findings of NSIP. To date, the gold standard for the diagnosis of IIPs, as is the case for the other ILDs, is represented by the multidisciplinary approach that can improve the pathology management and promote an appropriate therapeutic treatment. Indeed, along with the multidisciplinary approach, a more comprehensive study of the molecular mechanisms underlying the specific IIPs as well as of the shared molecular mechanisms could support both the diagnosis and the prognosis of these patients as well as widen the perspectives for potential new target therapies.

The typical high-resolution computed tomography (HRCT) of different IIPs shows the UIP pattern where the main features are represented by a reticular pattern, traction bronchiectasis and honeycombing appearance. These imaging findings are predominantly located in the subpleural regions and in lower lobes. In particular, IPF, which is considered idiopathic (?) but has several risk factors, such as the gender and age, genetic susceptibility, environmental risks and cigarette smoking, shows in Panel A peripheral, subpleural and basal pattern of distribution characterized by reticular opacities, honeycombing, minimal ground-glass opacity and architectural distortion. NSIP that can be either idiopathic or associated with connective tissue diseases (e.g., scleroderma), hypersensitivity pneumonia and drug reactions, more often affecting women ranging from 40 to 50 years of age, shows (Panel B) from the HCRT extensive ground-glass opacity and the presence of traction bronchiectasis with subpleural sparing in fibrotic non-specific interstitial pneumonia. COP, mostly found as secondary to connective tissue diseases, drug-induced adverse pulmonary reactions, hypersensitivity pneumonia and infectious processes affecting patients between 50 and 70 years of age, shows a radiological pattern (Panel C) characterized by peripheral or peribronchial patchy consolidations as well as ground-glass opacities with a tendency to migrate. RB-ILD mostly affects smokers between 30 and 60 years of age without gender preference. The HRCT of RB-ILD patients (Panel D) reveals central and peripheral bronchial wall thickening, centrilobular nodules, and ground-glass opacities together in some case with centrilobular emphysema at the upper lobe. AIP affects patients between 50 and 60 years of age with unknown etiology. The HRCT (Panel E) shows ground-glass opacities and air space consolidation together with traction bronchiectasis. DIP, mostly affecting men between 30 and 50 years, is strongly related to cigarette smoking, occupational risk exposure or use of inhalation drugs. The HCRT (Panel F) shows bilateral ground-glass opacities with lower lobe predominance. LIP has been associated with several autoimmune disorders (e.g., Sjögren’s syndrome, rheumatoid arthritis, systemic lupus erythematosus), virus and genetic predisposition. The HCRT (Panel G) displays thin-walled cysts in a random distribution, along with ground-glass opacities, centrilobular and subpleural nodules and reticulonodular opacities.

## 3. Molecular and Cellular Key Players behind IIPs

Idiopathic pulmonary fibrosis—IPF—In normal lungs, when alveolar injuries occur there is a depletion of alveolar epithelial cells 1 (AECI) located at the interface, with vascular endothelium participating in alveolar gas exchange functions. During the depletion of AECI that leads to a loss of lung epithelium integrity, the alveolar epithelial cells 2 (AECII) that are normally deputed to the secretion of the pulmonary surfactant to maintain the surface tension acquire the capacity to proliferate and differentiate into AECI. Thus, the cell “emergency” role of AECII restores the alveolar epithelium once injured, retaining the biomechanics property of alveolar epithelium [51]. Thus, the aberrant reparative mechanism of AECs as a response to injury in IPF patients initiates the development and progression of disease. Here, the AECs, through the secretion of pro-fibrotic factors, orchestrate all the following processes that enable the disease progression in IPF alveolar lungs: fibroblast migration, proliferation, activation and differentiation into myofibroblasts with the deposition of exaggerated ECM and subsequently the distortion of the lung architecture [24]. Thus, for the crucial role that AECs play in the initiation and progression of the disease, the IPF has been considered an “epithelium driver disease”. Here, several molecular pathways and mechanisms have been identified in lung IPF epithelium that contribute to the progression of the disease. Among these, pro-fibrotic mediators such as transforming growth factor beta-1 (TGF-β1), platelet-derived growth factor (PDGF), tumor necrosis factor (TNF), endothelin-1, connective tissue growth factor (CTGF), osteopontin, and CXC chemokine ligand 12 (CXCL12) [52], which are overexpressed in the AECs of IPF lungs, modulate both the progression of fibrosis and the aberrant extracellular matrix deposition [24]. Besides the presence of several molecular mechanisms responsible for the progression of IPF, the molecular driver in IPF progression can be considered TGFβ, which triggers the proliferation, migration and fibroblast favoring of transdifferentiation into myofibroblasts, culminating in the deposition of aberrant extracellular matrix [53]. The molecular signaling down-stream of the TGFβ receptor complex activation includes the canonical (mothers against decapentaplegic SMAD2 and 3) [54] and non-canonical signaling cascades (phosphoinositide 3-kinase PI3K, mitogen-activated protein kinase MEK, mechanistic target of rapamycin mTOR, etc.) [55] that modulate the transcription of profibrotic mediators, growth factors, microRNAs, and ECM proteins [56]. Indeed, mitogen-activated protein kinases (MAPKs), extracellular signal-regulated kinase (ERK), c-jun N-terminal kinase (JNK), and p38 kinase (p38 MAPK), which modulate cell proliferation, apoptosis, cell survival and cell motility [57], have been demonstrated to play an important role in the development of IPF. Initially, it was demonstrated that the MAPKs ERK JNK and p38 MAPK were both activated in lung tissues from patients with IPF compared with control lung parenchyma [58]. In particular, ERK activation in lung epithelial and endothelial cells was decreased as fibrosis progressed, while JNK activation was increased in the same cells and p38 MAPK was activated in lung smooth muscle cells, fibroblasts, endothelial and epithelial cells at the intermediate stage of fibrosis. Then, targeting the MEK pathway in fibrotic lung disease through the chemical inhibition of MEK prevented the progression of established lung fibrosis in the overexpressed TGF-α mouse model [58]. Recently, Kawara and collaborators demonstrated that Spred2 deficiency that represents an inhibitor of the rat sarcoma virus (Ras)/Rapidly Accelerated Fibrosarcoma (Raf)/MEK/ERK pathway involved in cell proliferation and inflammation, increasing the proliferation of lung epithelial cells, ameliorates pulmonary fibrosis induced by bleomycin [59]. Indeed, one of the most abundant heat shock proteins (HSPs), HSP90, that belongs to a large family of co-chaperones involved in the maintenance of proper cell protein homeostasis (proteostasis) [60], is increased in patients with IPF [61] and has recently been studied as both a biomarker of lung fibrosis and as a potential therapeutical target [62]. In particular, HSP90 activation in IPF followed by inflammation increases the synthesis and aberrant deposition of extracellular matrix proteins and collagen. Thus, it was demonstrated that the inhibition of HSP90 leads to the blocking of the TGF-β signaling pathway that is mainly responsible for disease progression [63,64]. 

Therefore, the cross-talk with TGFβ1 signaling and overexpressed pathways in epithelial cells of IPF patients such as Wnt/β-catenin and Sonic hedgehog (Shh) has been demonstrated in different works [65,66]. In particular, the activation of Wnt/β-catenin might stimulate the epithelial mesenchymal transition and activation of the myofibroblast [67], anti-apoptotic and pro-fibrotic phenotypes in lung fibroblasts of IPF patients [68] as well as increase AECII senescence, contributing to impaired lung repair [69]. The Shh pathway was found to be overexpressed and active in lung resident mesenchymal stromal cells LR-MSCs with the myofibroblastic phenotype, where it represented a key regulator of LR-MSCs to myofibroblast transition in IPF lungs. Indeed, the downregulation of the Shh–Wnt pathway inhibited myofibroblast differentiation from LR-MSCs ameliorating pulmonary fibrotic lesions [70]. Other pathways involved in the progression of IPF are represented both by platelet-derived growth factor (PDGF) that contributes to fibroblast activation and connective-tissue growth factor (CTGF) that, through the activation of fibroblasts and the TGF-β pathway, allows for the progression of fibrosis [71]. Indeed, C-X-C Motif Chemokine Ligand 12 (CXCL12) acting on CXCR4 is activated during the acute exacerbation of IPF and other ILDs [72], while the matrix metalloproteases (MMPs), such as MMP1 and MMP7, exploit their role in the progression of IPF increasing epithelial cell proliferation and preventing their apoptosis [24,73]. Furthermore, the Hyppo pathway plays a role in the molecular scenario of IPF through the activation of yes-associated protein (YAP) and the transcriptional coactivator with PSD-95/Discs large/ZO-1 homologous (PDZ)-binding motif (TAZ). In particular, YAP/TAZ that are master regulators of cell mechanical force are activated both in IPF lung fibroblasts where they secrete profibrotic factors and ECM proteins and in epithelial cells where they compromise cell polarity increasing ECM stiffness [74]. Here, in the lung of IPF patients, YAP/TAZ together with TGF-β, Wnt, and PI3K signaling pathways increase the proliferation and migration of epithelial cells and therefore the progression of the disease [75,76]. Indeed, the Hyppo pathway through TAZ modulates the lung repair mechanism since it is involved in the differentiation of AECII and AECI, contributing to preserving the lung epithelial integrity and thus the lung biomechanics properties [77]. To this purpose, the conditional deletion of TAZ in AECII reduced AECI regeneration during recovery after injury, causing lung fibrosis in a bleomycin-induced lung injury model. Hence, in this enriched molecular scenario responsible for the onset/progression of IPF, the resident lung fibroblasts represent the key effector cells since through the TGF-β action they can proliferate and differentiate into myofibroblasts, leading to lung fibrosis [78]. Indeed, resident lung fibroblasts expressing collagen and mesenchymal proteins, such as Vimentin and α-smooth muscle actin (α-SMA), trigger the deposition of aberrant ECM, leading to an increase in the lung parenchyma rigidity and stiffness and impairment of gas alveolar gas exchange [79]. According to recent studies based on lineage tracing experiments, the fibroblasts able to differentiate into myofibroblasts in IPF lung were PDGFRα-expressing interstitial fibroblasts and/or lipofibroblasts [80]. Interstitial lung fibroblasts localized in the lung interstitium closed to AECs and vascular endothelium play a crucial role in the maintenance of alveolar gas exchange, while lipofibroblasts whose specific markers are Sca-1, CD248 [81], are characterized by the presence of neutral lipids and are also localized close to the AECs, where they are involved in alveolar maturation and surfactant production as well as FGF10 secretion [36]. To date, besides the role of lipofibroblasts and interstitial fibroblasts, other cell populations involved in the transdifferentiation into myofibroblasts are represented by pericytes localized within the capillary basement membrane [37] mesothelial cells [38] and resident lung mesenchymal progenitors [39]. Indeed, in addition to the above-mentioned cellular population, a pivotal role of alveolar macrophages (AMs) in the immunopathogenesis of IPF and its fibrotic progression have been reported. In particular, during IPF progression, both the resident and recruited alveolar AMs undergo M2-like macrophages polarization [82], upregulating CCL-18 that triggers excessive collagen deposition from lung resident fibroblasts [83], TLR2 pathways [84], TGF-β1 [85], and apoptosis resistance [86]. Finally, Yamashita and collaborators showed that the macrophage profiles in idiopathic ILDs were characterized by a decrease in the CD163+ macrophages that correlate with poor prognoses [87].

Although there is no treatment available to cure the IPF, the two drugs currently in use, pirfenidone and nintedanib, have been demonstrated to slow disease progression, ameliorating the survival without a real improvement in quality of life and with some tolerability issues [13,88]. In particular, pirfenidone is a synthetic compound with anti-inflammatory and anti-fibrotic action downregulating pro-fibrotic growth factors including TGF-β [89]. Nintedanib is a synthetic compound acting as an inhibitor of tyrosine kinase receptors VEGFR1-3, FRGFR1-3 and PDGFRa-b involved in the proliferation and migration of lung fibroblasts as well as the differentiation of the key mechanisms of fibroblasts and myofibroblasts in IPF pathogenesis [90]. Indeed, many experimental treatments for IPF are currently available that specifically target the main molecular pathways responsible for the disease progression that we described above. Among them, the Pamrevlumab is a fully human monoclonal antibody against CTGF with an anti-fibrotic effect in a Phase III randomized, double-blind, placebo-controlled trial [91,92]. Indeed, the rhPTX-2/PRM-151 that modulates the TGF-β1 signaling in alveolar macrophages and their mediators is a Phase III, randomized, double-blind, placebo-controlled trial (ClinicalTrials.gov Identifier: NCT04552899), while the KD025/SLx-2119 that is a ROCK2 inhibitor with an anti-fibrotic effect is a Phase II open label trial [93] (ClinicalTrials.gov Identifier: NCT04552899). Furthermore, among the inhaled treatments that have been poorly studied in IPF [94], there is TRK-250, which suppresses the expression of TGF-β1 at the gene expression level in a Phase I, randomized, double-blind and placebo-controlled trial (ClinicalTrials.gov Identifier: NCT03727802). Finally, mesenchymal stem cells (MSCs) that are isolated from different sources have been demonstrated to play anti-fibrotic, anti-inflammatory and immunomodulatory effects [95,96]. Specifically, pre-clinical studies show that the administration of MSCs leads to the downregulation of TGF-β signaling, reducing the extent of fibrotic lesions and lung collagen content [97]. To date, the MSC administration has been studied in the Phase I trial assessing their safety [98], while there are issues and concerns concerning their therapeutic role that still need to be clarified [99].

## 4. Non-Specific Interstitial Pneumonia (NSIP) and Cryptogenic Organizing Pneumonia (COP)

NSIP that can display clinical similarities to IPF, as we have discussed in the previous section, presents a better outcome compared to the last one. Although the molecular mechanism of NSIP is unknown compared to the most studied IPF, the apoptosis of AECII that plays a crucial role in the IPF [100] has also been found in NSIP patients [101]. Indeed, the surfactant protein C (SP-C) gene mutations have been found mostly in familial forms of both IPF and NSIP, together with an increase in the production of misfolded proteins, leading to proteasome inhibition and endoplasmic reticulum (ER) stress that culminate in AECII depletion [102]. Nevertheless, the molecular mechanism that in NSIP leads to ER stress and AECII apoptosis is still unknown, and so is the specific pathway that allows for the discrimination of NSIP from the other IIPs. To this purpose, Korfei et al. performed a comparative proteomic analysis of peripheral lung tissue from 14 patients with sporadic IPF, 8 patients with fibrotic NSIP and 10 organ donors as controls. According to their results, the proteomic profiles of IPF and fNSIP were quite similar. In particular, in their proteome profile, both IPF and fNSIP shared the overexpression of proteins involved in the ER stress pathway, as well as the downregulation of antiapoptotic factors and antifibrotic molecules. In contrast to IPF, the NSIP proteome profile shows an overexpression of antioxidant proteins as a reparative mechanism against ROS and misfolded protein carbonyls that might explain the better outcome and survival in patients with NSIP compared to IPF [103]. Recently, Yamashita and collaborators found an increase in CD68+ macrophage density in UIPs such as NSIP, and COP, in mild and severe fibrotic lesions of IPF, while they were undetectable in the IPF fibroblastic foci. Intriguingly, they found an increase in the CD163+/CD68+ ratio in NSIP and COP compared to IPF characterized by a significant decrease in the CD163+/CD68+ macrophage density ratio [87]. Indeed, it was recently demonstrated that the presence of CD20+ B lymphocytes infiltrating the lung parenchyma correlates with poor prognosis in the fibrosing NSIP [104]. Furthermore, a comprehensive gene expression profiling was performed using microarray analysis on explanted lungs that identified distinct transcriptional profiles and differentially expressed genes both in IPF and NSIP. In particular, they found an enrichment in genes related to immune reaction T-cell response and the recruitment of leukocytes into the lung compartment in NSIP, while the IPF transcriptomic profile was enriched with senescence, epithelial-to-mesenchymal transition, myofibroblast differentiation and collagen deposition genes [105]. Despite the recent immunophenotyping of NSIP and COP patients, we are far from the identification of molecular signature that unequivocally might discriminate either the NSIP or COP from the other IIPs, and further challenging works in this direction are needed. The therapeutic management of NSIP patients is based on the use of immunosuppressive or immunomodulatory agents that might lead to disease regression or stabilization, such as corticosteroids, cyclophosphamide, azathioprine, mycophenolate mofetil, and rituximab, which are chimeric anti-CD20 B cell-depleting monoclonal antibodies. Finally, a recent phase II trial studied the efficacy and safety of pirfenidone in fibrotic unclassifiable disease, some of which may include cases of NSIP [106,107].

## 5. Acute Interstitial Pneumonia (AIP)

Acute interstitial pneumonia (formerly the Hamman–Rich syndrome) is a rare IIP that is characterized by rapid and progressive clinical course, respiratory failure and high mortality [108]. Here, the myofibroblasts, much like they do for IPF, play a crucial role in the progression of diffused alveolar damage (DAD), which represents a key characteristic of AIP. To this purpose, Li et al. stated that the myofibroblast may origin from AECs that undergo the epithelial–mesenchymal transition (EMT) [109], a cellular process that has been an object of debate for a long time [110]. Recently, Chen and collaborators demonstrated that thymocyte differentiation antigen-1 (THY1), also known as CD90, modulated lung fibroblast proliferation and fibrogenic signaling, namely, the progression of AIP. In this study, the overexpression of THY1 led to a significant decrease in the expression level of Matrix Metallopeptidase 2 (MMP-2), Occludin, α-SMA, Vimentin and β-catenin, and the extent of the β-catenin phosphorylation of pulmonary fibrosis and therefore the inactivation of the WNT signaling pathway. Thus, cell proliferation was inhibited and apoptosis was enhanced in lung fibroblasts overexpressing the THY1 vector [111]. Therefore, it is of great importance to further understand the mechanisms and pathogenesis of AIP to develop more effective therapeutic strategies. To this purpose, current therapeutic treatments for AIP besides the supportive care are represented by glucocorticoids and immunosuppressive therapy [108].

## 6. Desquamative Interstitial Pneumonia (DIP) and Respiratory Bronchiolitis-Associated Interstitial Lung Disease (RB-ILD)

Although DIP and RB-ILD belong to idiopathic interstitial pneumonia, during the last years, cigarette smoking has been implicated in the pathogenesis of diffuse interstitial lung disease DILD as RB-ILD and DIP. Thus, although the molecular and cellular mechanism behind the onset and progression of DIP and RB-ILD is unknown, it can be partially described as the effect of cigarette smoking in lung patients. Specifically, it has been demonstrated that nicotine, which represents the addictive substances in cigarette smoking, represents a crucial initiator of the pro-fibrotic machinery since it triggers the epithelial and cellular damage, it increases the production and the secretion of TGF-β1, it stimulates the recruitment of inflammatory cells and the production of reactive oxygen species and it promotes the production and deposition of excessive collagen and therefore of the extracellular matrix [112]. DIP and RB-ILD are both considered inflammatory disorders, based on histologic studies, characterized by diffuse interstitial inflammation, peribronchiole accumulation of macrophages, and fibrosis; BAL fluid from both patients with RB-ILD and with DIP is hypercellular, with an increase in the number of macrophages, neutrophils and lymphocytes greater than non-smoking control subjects [113]. BAL and serum from subjects with RB-ILD and DIP have a higher expression level of C-C chemokine ligand 18, a cytokine produced by activated alveolar macrophages [114]. Indeed, both the aberrant telomere shortening and compromised mechanism of autophagy are involved in the pathogenesis of both smoking-related ILD and IPF where cigarette smoking represents a strong risk factor. Here, the aberrant mechanism of telomere shortening affecting the AECs of IPF lung decreases the apoptosis process though the p53 mechanism increasing cellular senescence and compromising the lung epithelium regeneration [115]. In IPF, the aberrant mechanism of telomere shortening can be due to either the genetic mutation affecting the telomerase, the protein responsible for telomere shortening, or cigarette smoking [116,117]. Indeed, it has been found that in IPF lungs, the increased expression level of TGF-β1 causes the suppression of autophagy, leading to dysfunctional mitochondria and aggregated proteins [118,119]. Finally, patients with RB-ILD associated with cigarette smoking that do not ameliorate after smoking cessation may be treated with steroids. Treatment for DIP patients consists of corticosteroid therapy, which may cause stabilization or improvement of the disease, while in some patients the progression of disease occurs besides this treatment [120].

## 7. Lymphocytic Interstitial Pneumonia (LIP)

Lymphocytic interstitial pneumonia (LIP) is an interstitial lung disease encompassing the spectrum of benign pulmonary lymphoproliferative disorder that involves an inflammatory pulmonary reaction based on cellular expansion and infiltration of the interstitium by T and B lymphocytes and plasma cells [121]. Furthermore, LIP is associated in most of the cases with autoimmune disease. The key pathologic feature of LIP is the presence of interstitial lymphocytic infiltrates, which extend to interlobular and alveolar septa [122]. Thus, the key cellular players of LIP are the lymphocytes together with plasma cells, immunoblasts and macrophages. Indeed, in the lung niche of LIP patients, there are T cells that express CD3 mostly located in the interstitium, while B lymphocytes expressing CD20 characterize half of the patients with the idiopathic and connective tissue-associated forms of LIP [123]. Thus, we could speculate that the molecular mechanism characterizing the progression of LIP might describe the migration and invasion process of the effector T cell during the inflammatory damage. To this purpose, the molecular mechanisms that modulate the effector T cells’ migration towards the interstitial space of inflamed lungs are incompletely understood. Recently, Mrass et al. studied the T-cell migration in a mouse model of acute lung injury though two-photon imaging of lung tissue. Here, after computational analysis, they found that T cells through the activation of Rho-associated protein kinase (ROCK) migrated both with high speed and with straight motion, guided by lung-associated vasculature [124]. Then, aberrant expression of HLA-DR (human leukocyte antigen-D related) in non-immune cells that led to an increased expression level of TGF-β correlating with LIP pathogenesis was reported in some patients. Finally, LIP associated to Sjögren’s syndrome may be treated with corticosteroids although the response is unpredictable. Other therapeutic options are represented by abatacept, a CTLA-4 monoclonal that specifically targets the T cells decreasing lymphocytic foci and local T cells. Indeed, the combination of rituximab and belimumab is a recent therapeutic approach that aims to target B-cells (trial NCT02631538) [125].

## 8. Conclusions and Perspectives

The diagnosis and classification of different IIPs is challenging, thus requiring a multidisciplinary debate to bring it to a conclusion and then to define an effective treatment. Given the similarity of the clinical and pathological findings elapsing some of the IIPs, together with the complexity of the differential diagnoses, a molecular signature and a defined cellular phenotype underpinning single disease onset and progression are needed. It has been observed from recent studies that IIPs having similar risk factors (cigarette smoking, presence of autoimmune diseases) also share the activation of similar molecular pathways, such as oxidative stress, ER stress, the aberrant autophagy process and lymphocytes infiltration, respectively (Figure 2). Notwithstanding this, we are still far from obtaining specific molecular and cellular profiles in all individual IIPs, including IPF. For this reason, further studies with this aim are of fundamental importance to identifying molecular markers and to apply efficient target therapies.

Injuries cause the depletion of AECI that in IPF lung epithelium is not compensated with the differentiation of AECII in AECI. Together with the aberrant repair mechanisms, the activated AECs trigger the secretion of pro-fibrotic molecules, coagulant and cytokines that activate several cells from different sources: the interstitial lung fibroblast, the resident mesenchymal stromal cells, the pericytes and the lipofibroblasts. Then, these different cell types in IPF lungs differentiate into myofibroblasts with the consequent deposition of extracellular matrix proteins and an overall lung structural rigidity together with impaired alveolar gas exchange. The NSIP and COP are characterized by an increase in the CD163+ macrophages and CD20+lymphocytes, misfolded proteins, ER stress and mitochondria dysfunction with excessive production of ROS. In AIP, CD90 leads to interstitial fibroblast activation and proliferation, differentiation into myofibroblasts as well as alveolar epithelial cells according to one of the most debated cellular processes in the field of fibrotic disease: the EMT. In DIP and RB-ILD, the increased secretion of tgf-β1 leads to the deposition of collagen and extracellular matrix proteins together with ROS excessive production. Along with this response, there is also an increase in the inflammatory cells, such as alveolar macrophages and lymphocytes. LIP is mainly characterized by an inflammatory response with infiltration of alveolar macrophages, together with B-lymphocytes CD3+ and T-lymphocytes CD20+ that are able to migrate through the ROCK pathway.

## Figures and Tables

**Figure 1 ijms-22-08952-f001:**
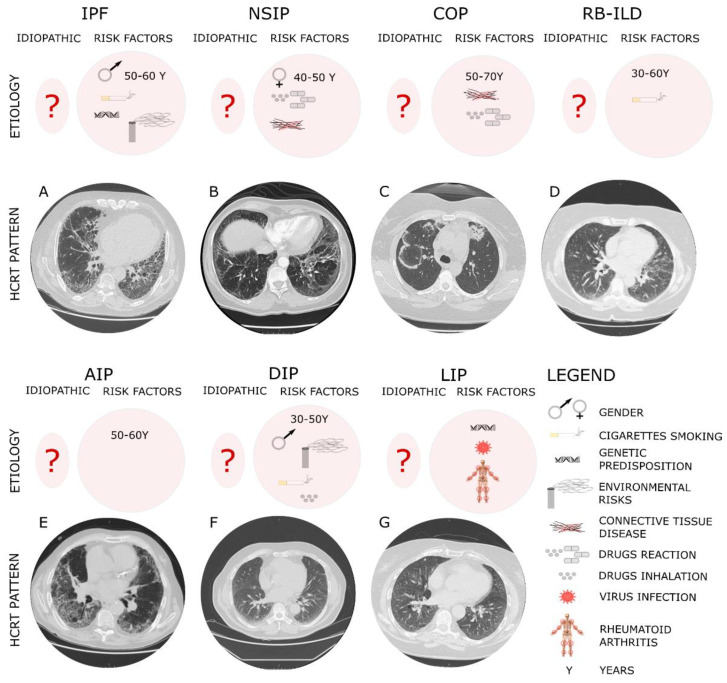
The diagnostic and etiologic characteristic of different IIPs.

**Figure 2 ijms-22-08952-f002:**
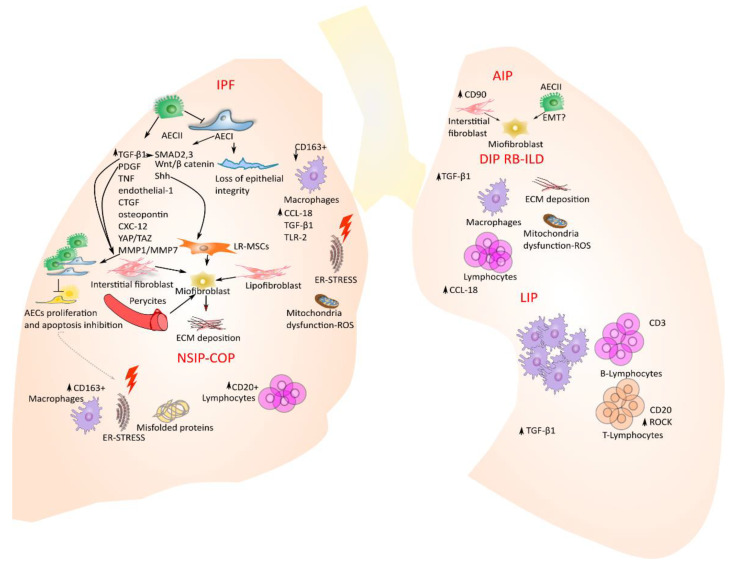
Molecular key players and signaling pathway characterizing the lung niche of IIP patients. Arrows indicate the molecular activation signaling leading to myofibroblasts activation and/or fibrotic disease progression process.

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
