# Peer review of "Fibrotic Idiopathic Interstitial Lung Disease: The Molecular and Cellular Key Players"

_ijms, 2021, doi:10.3390/ijms22168952_

Round 1

Reviewer 1 Report

In the manuscript "Fibrotic idiopathic interstitial lung disease: the molecular and cellular key players" the authors have covered a full range of related diseases.

There are a few remarks to help the authors improve the review. 

  1. In paragraph 3 authors discuss the canonic SMAD2/3 TGFb signaling in detail. However, the non-canonic MAPK/ERK pathway should be also mentioned as an also very important pathway, which regulates the severity of pulmonary fibrosis and differences between men and women. In addition, a discussion about HSP90 as an important profibrotic biomarker should be added.  
  2. I would suggest to the authors adding a couple of sentences after each paragraph to discuss potential experimental treatment,  based on the aforementoined molecular players.
  3. There are some minor spelling and punctuation errors in the text. Also, the reference should be before the dot at the end of the sentence but not after.

Author Response

On behalf of all the Authors we would like to thank the Editors and all the Reviewers for their valuable time and useful contribution to our Manuscript “Fibrotic idiopathic interstitial lung disease: the molecular and cellular key players”. We strongly appreciate the suggestions and inputs they have given that will definitely improve our manuscript updating the molecular and cellular key players in the idiopathic interstitial lung disease. Following their suggestions we have enriched the manuscript to better and clearly explained concepts. Please find below the details.

Reviewer 1:

1)In paragraph 3 authors discuss the canonic SMAD2/3 TGFb signaling in detail. However, the non-canonic MAPK/ERK pathway should be also mentioned as an also very important pathway, which regulates the severity of pulmonary fibrosis and differences between men and women. In addition, a discussion about HSP90 as an important profibrotic biomarker should be added.  

We thank the Reviewer for the suggestion. We have described the role of the non-canonic MAPK/ERK pathway in the development of pulmonary fibrosis in Paragraph 3, pag 7 Line 277

“Indeed, Mitogen activated protein kinases (MAPKs), extracellular signal-regulated kinase (ERK), c-jun N-terminal kinase (JNK), and p38 kinase (p38 MAPK), that modulate cell proliferation, apoptosis, cell survival and cell motility[57] have been demonstrated to play an important role in the development of IPF. Initially it has been demonstrated that MAPKs ERK JNK and p38 MAPK were all activated in lung tissues from patients with IPF compared with control lung parenchyma[58]. In particular, ERK activation in lung epithelial and endothelial cells was decreased as fibrosis progressed while JNK activation was increased in the same cells and p38 MAPK was activated in lung smooth muscle cells, fibroblasts, endothelial and epithelial cells at the intermediate stage of fibrosis. Then, targeting MEK pathway in fibrotic lung disease through chemical inhibition of MEK prevented the progression of established lung fibrosis in the overexpressed TGF-α mouse model.[59] Recently, Kawara and collaborators demonstrated that Spred2 deficiency that represents an inhibitor of the Ras/Raf/MEK/ ERK pathway involved in cell proliferation and inflammation, increasing the proliferation of lung epithelial cells ameliorates pulmonary fbrosis induced by bleomycin [60]. Indeed, HSP90, one of the most abundant Heat shock proteins (HSPs) that are a large family of co-chaperone involved in the maintenance of proper cell protein homeostasis (proteostasis)[61], is increased in patients with IPF [62] and has recently been studied as both a biomarker of lung fibrosis and as potential therapeutical target [63]. In particular, HSP90 activation in IPF following by inflammation, increase the synthesis and aberrant deposition of extracellular matrix proteins and collagen. Thus, it has been demonstrated that inhibition of HSP90 leads to the blocking of the TGF-β signaling pathway the main responsible of disease progression[64][65].

2)I would suggest to the authors adding a couple of sentences after each paragraph to discuss potential experimental treatment, based on the aforementoined molecular players.

We add after each paragraph that describe the main molecular pathway responsible for the progression of the disease, the potential therapy available/in development targeting that specific pathway described. Details can be found below.

IPF Pag 8 Line 354

Although there is no treatment available to cure the IPF, the two drugs currently in use, Pirfenidone and Nintedanib have been demonstrated to slow disease progression ameliorating the survival without a real improvement of quality of life and with some tolerability issues. [89][90] In particular, Pirfenidone is a synthetic compound with anti-inflammatory and anti-fibrotic action down-regulating pro-fibrotic growth factors including TGF-β. [91] Nintedanib is a synthetic compound acting as inhibitor of tyrosine kinase receptors VEGFR1-3, FRGFR1-3 and PDGFRa-b involved in the proliferation and migration of lung fibroblasts, and differentiation of fibroblasts to myofibroblasts key mechanisms of IPF pathogenesis. [92]Indeed, many experimental treatments for IPF are currently available that specifically target the main molecular pathways responsible for the disease progression that we described above. Among them the Pamrevlumab is a fully human monoclonal antibody against CTGF with anti-fibrotic effect in a Phase III randomized, double-blind, placebo-controlled trial [93] [94]. Indeed, the rhPTX-2/PRM-151 that modulates the TGF-β1 signaling in alveolar macrophages and their mediators is a Phase III, randomized, double-blind, placebo-controlled trial (ClinicalTrials.gov Identifier: NCT04552899), while the KD025/SLx-2119 that is a ROCK2 inhibitor with anti-fibrotic effect is a Phase II open label trial [95](ClinicalTrials.gov Identifier: NCT04552899). Furthermore, among the inhaled treatments that have been poorly studied in IPF, [96] there is TRK-250 that suppress the expression of TGF-β1 at gene expression level in a Phase I, randomized, double-blind and placebo-controlled trial (ClinicalTrials.gov Identifier: NCT03727802). Finally, Mesenchymal Stem Cells (MSCs) that are isolated from different sources have been demonstrated to play an anti-fibrotic, anti-inflammatory and immunomodulatory effects. [97], [98]. Specifically, pre-clinical studies show that administration of MSCs lead to downregulation of TGF-β signaling, reducing the extent of fibrotic lesions and lung collagen content. [99] To date, the MSCs administration have been studied in Phase I trial assessing their safety [100] while there are issues and concerns on their therapeutic role that still need to be clarified. [101]

NSIP Pag 9 Line 390

Therapeutic management of NSIP patients are based on the use of immunosuppressive or immunomodulatory agents that might lead to disease regression or stabilization such as Corticosteroids, Cyclophosphamide, Azathioprine, Mycophenolate Mofetil, and Rituximab that is chimeric anti-CD20 B cell depleting monoclonal antibody. Finally, a recent phase II trial studied the efficacy and safety of Pirfenidone in fibrotic unclassifiable disease, some of which may include cases of NSIP. [108][109]

AIP Pag 9 Line 407

Current therapeutic treatments for AIP besides the supportive care are represented by glucocorticoids and immunosuppressive therapy.[114]

DIP and RB-ILD, Pag 9 Line 435

Patients with RB-ILD associated with cigarette smoking that do not ameliorate after smoking cessation may be treated with steroids. Treatment for DIP patients consists of corticosteroid therapy, which may cause stabilisation or improvement of the disease, while in some patients the progression of disease occurs besides this treatment. [123]

LIP Pag 10 Line 458

LIP associated to Sjögren's syndrome may be treated with corticosteroids although the response is unpredictable. Others therapeutic options are represented by abatacept a CTLA-4 monoclonal that specifically targets the T-cells decreasing lymphocytic foci and local T-cells. Indeed, the combination of rituximab and belimumab is a recent therapeutic approach that aims to target B-cells (trial NCT02631538).[128]

There are some minor spelling and punctuation errors in the text. Also, the reference should be before the dot at the end of the sentence but not after.

Agreed. We have correct accordingly.

Reviewer 2 Report

The paper is overall interesting and suitable to the journal audience. My only concer regards the translational potential of all the data presented and discussed. I suggest to add a section dedicated to the therapeutic strategies in study and development to target the molecular pathways described. 

Author Response

On behalf of all the Authors we would like to thank the Editors and all the Reviewers for their valuable time and useful contribution to our Manuscript “Fibrotic idiopathic interstitial lung disease: the molecular and cellular key players”. We strongly appreciate the suggestions and inputs they have given that will definitely improve our manuscript updating the molecular and cellular key players in the idiopathic interstitial lung disease. Following their suggestions we have enriched the manuscript to better and clearly explained concepts. Please find below the details.

Reviewer 2:

The paper is overall interesting and suitable to the journal audience. My only concern regards the translational potential of all the data presented and discussed. I suggest to add a section dedicated to the therapeutic strategies in study and development to target the molecular pathways described. 

We thank the Reviewer and we have added further details on the therapeutic strategies available and the experimental ones for the ILDs after each section dedicated to the specific ILD subtype in order to correlate them to the molecular pathway described in the same section. Further details can be found above in the answer 2) to the Reviewer 1.
